# Marine Organisms from the Yucatan Peninsula (Mexico) as a Potential Natural Source of Antibacterial Compounds

**DOI:** 10.3390/md18070369

**Published:** 2020-07-18

**Authors:** Dawrin Pech-Puch, Mar Pérez-Povedano, Patricia Gómez, Marta Martínez-Guitián, Cristina Lasarte-Monterrubio, Juan Carlos Vázquez-Ucha, María Lourdes Novoa-Olmedo, Sergio Guillén-Hernández, Harold Villegas-Hernández, Germán Bou, Jaime Rodríguez, Alejandro Beceiro, Carlos Jiménez

**Affiliations:** 1Centro de Investigacións Científicas Avanzadas (CICA) e Departmento de Química, Facultade de Ciencias, AE CICA-INIBIC, Universidade da Coruña, 15071 A Coruña, Spain; dawrin.j.pech@udc.es (D.P.-P.); perezpovedanomaranabel@gmail.com (M.P.-P.); 2Unidad Académica de Ecología y Biodiversidad Acuática, Instituto de Ciencias del Mar y Limnología, Universidad Nacional Autónoma de México, CDMX 04510, Mexico; patricia@cmarl.unam.mx; 3Servicio de Microbioloxía, Instituto de Investigación Biomédica AE CICA-INIBIC, Complexo Hospitalario Universitario A Coruña, 15006 A Coruña, Spain; m.martinez.guitian@gmail.com (M.M.-G.); crlasarm@gmail.com (C.L.-M.); juan.vazquez@udc.es (J.C.V.-U.); marianovoa@usb.ve (M.L.N.-O.); German.Bou.Arevalo@sergas.es (G.B.); Alejandro.Beceiro.Casas@sergas.es (A.B.); 4Departamento de Biología Marina, Campus de Ciencias Biológicas y Agropecuarias, Universidad Autónoma de Yucatán, Carretera Mérida-Xmatkuil Km. 15.5, Mérida C.P. 97000, Yucatán, Mexico; ghernand@correo.uady.mx (S.G.-H.); harold.villegas@correo.uady.mx (H.V.-H.)

**Keywords:** sponges, ascidians, antimicrobial, multidrug-resistance, Yucatan Peninsula

## Abstract

A total of 51 sponges (Porifera) and 13 ascidians (Chordata) were collected on the coast of the Yucatan Peninsula (Mexico) and extracted with organic solvents. The resulting extracts were screened for antibacterial activity against four multidrug-resistant (MDR) bacterial pathogens: the Gram-negative *Acinetobacter baumannii*, *Klebsiella pneumoniae*, and *Pseudomonas aeruginosa* and the Gram-positive *Staphylococcus aureus*. The minimum inhibitory concentrations (MICs) of the organic extracts of each marine organism were determined using a broth microdilution assay. Extracts of eight of the species, in particular the *Agelas citrina* and *Haliclona (Rhizoniera) curacaoensis*, displayed activity against some of the pathogens tested. Some of the extracts showed similar MIC values to known antibiotics such as penicillins and aminoglycosides. This study is the first to carry out antimicrobial screening of extracts of marine sponges and ascidians collected from the Yucatan Peninsula. Bioassay-guided fractionation of the active extracts from the sponges *Amphimedon compressa* and *A. citrina* displayed, as a preliminary result, that an inseparable mixture of halitoxins and amphitoxins and (-)-agelasine B, respectively, are the major compounds responsible for their corresponding antibacterial activities. This is the first report of the antimicrobial activity of halitoxins and amphitoxins against major multidrug-resistant human pathogens. The promising antibacterial activities detected in this study indicate the coast of Yucatan Peninsula as a potential source of a great variety of marine organisms worthy of further research.

## 1. Introduction

The Yucatan Peninsula in Mexico is the area where the great Mayan culture became established and developed. The Maya people had extensive knowledge about the natural resources available and they maintained a long tradition of using the flora and fauna from the region to meet their basic needs in relation to food, health, and housing. In a recent large ethnobiological study of contemporary Mayan culture [1], is reported that some 145 of the estimated 2300 vascular plants known to grow in the region were used as food sources, while another 680 are used for medicinal purposes. The same study refers to different applications of various wildlife species: 81 terrestrial vertebrates, mainly reptiles, birds, and mammals, were used as food sources, and at least 40 wild vertebrates were used for medicinal purposes. More specifically, in relation to marine resources, 62 teleostean, 3 elasmobranch, 4 molluscan, and 4 crustacean species were commercially harvested [1].

The discovery of the remains of toxic marine animals, such as stingrays, puffer fishes, cone snails, sponges, and corals, in the caches of archaeological sites of the ancient Mayan civilization indicated that the venoms or toxins of these organisms may have been used for specific purposes, e.g., to induce pain or in ritual contexts [1]. Sponges were also used in ancient Greek civilization for medical and pharmacological purposes because of their absorbent, compressive, and sterile properties [2]. However, direct uses for marine organisms in Mayan culture are not known in many cases. 

Antimicrobial resistance (AMR) has become a global health emergency and is placing the great achievements made with the discovery of antibiotics in the 20th century at risk. The use and development of antibiotics did not take into account the rapid evolution of the bacteria being targeted, and antimicrobial resistance has emerged hand in hand with the routine clinical use of antibiotics. It has been estimated that AMR is responsible for more than 25,000 deaths a year in the European Union, and it has been predicted that this number could rise to more than 10 million deaths worldwide by 2050, overcoming pathologies such as cancer and heart disease [3]. In addition, AMR has been estimated to cause economic losses of 1.5 billion euros a year in European health systems. International organizations, such as the Food and Agriculture Organization (FAO), the World Organisation for Animal Health (OIE), and the World Health Organization (WHO), have established AMR as one of the main research objectives within the field of health in the next decade. This scenario indicates the need for extraordinary efforts to be made both from scientific and academic communities and from society in general, in the search for solutions to this global health problem.

The WHO has published a list of priority pathogens for R&D of new antibiotics. The list includes various species identified as of critical priority, such as *Acinetobacter baumannii*, the carbapenem-resistant *Enterobacterales* and *Pseudomonas aeruginosa*, and several species identified as being of high priority, such as *Staphylococcus aureus*, which is resistant to methicillin and vancomycin [4]. Hospital outbreaks caused by these pathogens have increased significantly in recent decades, causing serious health problems [5]. Carbapenems are last-line antibiotics used to treat multiresistant strains of these Gram-negative pathogens. However, resistance to these antimicrobials increased substantially in the first decade of this century [6]. Colistin is one of the few remaining antimicrobials that is effective against these Gram-negative pathogens, with resistance rates below 10%. Unfortunately, there are important drawbacks associated with the use of colistin, including high nephrotoxicity, poor distribution in cerebrospinal fluid and lung, and rapidly increasing resistance rates [7]. Under these circumstances, the need for new therapeutic options for the treatment of multi-resistant pathogen infections is indisputable. Most recently developed antimicrobials are derived from already known antibiotics. The discovery of new antimicrobials is closely followed by the development of resistance to these, and drugs with new mechanisms of action that overcome the current mechanisms of resistance are urgently required [8]. 

The oceans have become an important natural source of bioactive molecules, mainly because they cover a great proportion of Earth’s surface (more than 70%) and are highly diverse ecosystems. The distinctive physical and chemical conditions of the marine environment facilitate the production of molecules with unique structural, chemical, and biological characteristics that are not found in natural products isolated from terrestrial sources [9]. Most of the biological activities associated with marine natural products are cytotoxic and anticancer properties [10]. Natural products and their derivatives that can be related to molecules isolated from marine organisms and which have been approved as therapeutic agents belong to six structural types [11]. The alkaloid Yondelis® (trabectedin, ecteinascidin 743), a marine natural product isolated from the tunicate *Ecteinascidia turbinata* [12], and the antibody-drug conjugate (ADC) Adcetris® (brentuximab vedotin), a derivative of dolastatin 10, originally isolated from extracts of the sea hare *Dolabella auricularia* [13], represent two examples of anticancer agents approved by the US Food and Drug Administration (FDA). Moreover, a large number of marine-derived drug candidates are currently under evaluation in clinical trials. Porifera (sponges) and Cnidaria (soft corals), along with marine microorganisms, are currently the most productive sources of new natural products [14].

In tropical waters, species and their predators compete strongly for space, and marine organisms produce secondary metabolites as chemical defence mechanisms [15]. Mexico has been considered one of the three areas in the world with the highest terrestrial and marine biological diversity [16], and although the chemical diversity of the Mexican medicinal flora has long been investigated, yielding a wide variety of bioactive compounds [17], the chemical potential of the marine resources has not been widely explored [18]. The Mexican territory covers an area of 2,946,825 km^2^ and has 11,122 km of coastline, which extends from the Pacific Ocean to the Caribbean Sea and the Gulf of Mexico, including the Baja California peninsula (northwest) and the Yucatan peninsula (southeast), which hosts a rich marine flora and fauna [19]. More specifically, the Gulf of Mexico and the Caribbean Sea, which meet in the Yucatan channel, constitute two outstanding marine ecosystems. This particular geographical location promotes the existence of a high diversity and abundance of different marine species, which represent a potential source of bioactive compounds and food [20]. The Yucatan Peninsula occupies 17.4% of the coastline of Mexico and has a great biological diversity on the shore and in the ocean. Moreover, it has a considerable extension of shores and other biological zones, such as mangrove forests, tropical reefs, and protected natural areas (Celestún, Ría Lagartos, Dzilam de Bravo, and Alacranes Reef, among others), where the chemical properties of natural products remain untapped [19]. The few studies of marine organisms collected from the Yucatan peninsula have focused almost exclusively on describing the biological activities of the organic extracts, mainly antibacterial [20], antifungal [20], antioxidant [21], antiparasitic [22,23], ichtyotoxic [24], and cytotoxic [25] extracts. Most of the marine organisms that have been studied are algae, with fewer studies of cnidarians and holothuroids, and very few of sponges and tunicates [26,27]. 

In light of this background, we have begun a research project focusing on the chemistry and biological activity of sponges and ascidians from the Yucatan Peninsula [28]. In this article, we report the evaluation of the antibacterial activity of 64 organic extracts from marine invertebrate species collected along the coasts of the Yucatan Peninsula, selected on the basis of chemotaxonomic criteria. We prioritized those species of marine organisms belonging to genera or families from which compounds with unusual structures have been extracted or that have previously shown remarkable biological activity. Two of the active sponge extracts were further fractionated and the major compounds responsible for the antimicrobial activity were isolated.

## 2. Results and Discussion

### 2.1. Animal Material Studies

A total of 64 marine organisms (51 sponges and 13 ascidians) were collected from two different species-rich ecosystems in the Yucatan Peninsula in Mexico: a coral reef and a mangrove forest. For the collection, specimens belonging to genera or families, from which structurally novel compounds had previously been extracted and/or that displayed interesting biological activities, were prioritized (selected from SciFinder and Antimarin databases by applying chemotaxonomic criteria). Each organism was exhaustively extracted (three times) with a mixture of dichloromethane-methanol (1:1) to produce the corresponding organic extract. Aliquots of each extract were evaluated in antibacterial activity tests using the gold standard broth microdilution method. 

The bacterial pathogens *A. baumannii*, *P. aeruginosa*, *Klebsiella pneumoniae*, and *S. aureus* were used as test strains in this study because they form part of the ESKAPE group (an acronym that gather the names of six bacterial pathogens commonly associated with antimicrobial resistance), recognised as some of the most clinically relevant nosocomial pathogens. We have included, in this study, two strains of every bacterial species: one strain susceptible to classical antimicrobials and one a multidrug-resistant strain (Appendix A). There is an urgent and growing need for new classes of antibiotics to use in therapies that can ensure the level of infection control required in medical procedures [29].

In clinical microbiology, microbial susceptibility is defined by breakpoints, which are the lowest antibiotic concentrations that inhibit bacterial growth of any strains lacking resistance mechanisms [9]. Thus, the MICs are very valuable as a standardized measure of the antibiotic activity against bacteria. We established three categories of antibacterial activity to classify the marine extracts: high activity (minimum inhibitory concentration (MIC) ≤ 8 mg/L), intermediate activity (MIC 16–64 mg/L), and low or no activity (MIC ≥ 128 mg/L).

Of the 64 extracts tested, 9 displayed antimicrobial activity against one or more bacterial pathogens, according to the aforementioned classification. All extracts from tunicates were found to be inactive while the active extracts were derived from eight sponges: *Agelas citrina, Agelas dilatata, Agelas sceptrum, Aiolochroia crassa* (collected from two different locations: Mahahual in Quintana Roo state and Alacranes Reef in Yucatan state)*, Amphimedon compressa, Dysidea* sp., *Monanchora arbuscula*, and *Haliclona (Rhizoniera) curacaoensis* (Table 1). Three of these sponges showed antimicrobial activity against all the four bacterial strains tested. *A. citrina* showed the highest activity (MIC of 8 mg/L for the three Gram-negative species and a MIC of 0.5 mg/L for the Gram-positive *S. aureus*), followed by *A. compressa* with a MIC of 32 mg/L for the four bacterial species and finally *H. (Rhizoniera) curacaoensis*, with a MIC of 4–32 mg/L for the three Gram-negative and a MIC of 4 mg/L for *S. aureus*. The remaining five extracts displayed antimicrobial activity, with MIC values ranging from 16 to 128 mg/L. Thus, the *M. arbuscula* extract showed good activity against *S. aureus*, with a MIC of 16 mg/L, but was inactive against the three Gram-negative bacteria. The *Dysidea* sp. extract also displayed antibacterial activity, with MIC values of 16 and 32 mg/L for *A. baumannii* and *S. aureus*, respectively. The *A. dilatata* extract yielded MIC values of 32–128 mg/L for two of the Gram-negative bacteria and a MIC of 64 mg/L for *S. aureus*. For the extracts of *A. crassa* collected from two different sites, the MIC values ranged from 32–64 mg/L for *A. baumannii* and *S. aureus*. Finally, the *A. sceptrum* extract yielded a MIC value of 64 mg/L for *P. aeruginosa* (Table 1). 

Comparison of the MICs with those of known antibiotics revealed the high activity of some extracts. For example, penicillins (piperacillin and ticarcillin) and even carbapenems (meropenem) belonging to the β-lactam antibiotic group, for which MIC values of less than, respectively, 16 mg/L and of less than 8 mg/L for the causative strain, are used successfully to treat infections caused by Gram-negative pathogens. Aminoglycosides (e.g., amikacin), for which MICs of less than 16 mg/L have been determined, are generally clinically effective against Gram-negative species. For example, teicoplanin and linezolid are widely used to treat infections caused by *S. aureus* and are effective against strains with which MICs of less than 4 mg/L have been determined. Antibiotics such as fosfomycin, with low toxicity or adequate PK/PD parameters, reach high concentrations in serum and can be used to treat strains for which MICs < 32 mg/L have been determined [30].

The extract of *A. citrina* exhibited in vitro antibacterial activity similar to that of most antibiotics used in routine clinical practice, against both Gram-negative (MIC 8 mg/L) and Gram-positive (MIC 0.5 mg/L) pathogens. The extract of *H. (Rhizoniera) curacaoensis* also displayed high activity against *A. baumannii* (MIC of 4 mg/L), a pathogen considered one of the opportunistic pathogens most threatening to global health and for which limited therapeutic options are available. The high genetic plasticity of this organism allows it to adapt quickly to unfavourable contexts and readily develop antibiotic resistance. This pathogen has been cited by some authors as a paradigm of multidrug-resistance [31]. Very few antimicrobials display higher in vitro activity against *A. baumannii* than this extract, e.g., imipenem and colistin, which are used as last-line antibiotics to treat severe infections (susceptibility breakpoint ≤ 2 mg/L). The in vitro activity of *H. (Rhizoniera) curacaoensis* against *S. aureus* (MIC 4 mg/L), was similar to the susceptibility breakpoints reported for antimicrobials used to treat this pathogen, such as vancomycin and linezolid (susceptibility breakpoints of ≤2 and ≤4 mg/L, respectively). 

In addition to three of the extracts being active against all pathogenic species tested (*A. citrina*, *A. compressa* and *H. (Rhizoniera) curacaoensis*), the narrow-spectrum antibacterial activity displayed by another three extracts (*M. arbuscula*, *Dysidea* sp. and *A. sceptrum*) against a maximum of two of the four bacteria tested) is also worth highlighting (Table 1). Most antibiotics used to treat bacterial infections are active against multiple species showing relevant benefits for use in clinical settings. However, the use of broad-spectrum antibiotics has two main drawbacks: (i) selection for resistant, pathogenic, and non-pathogenic bacteria leads to selection for resistance genes that decrease the usefulness of that antibiotic; and (ii) the use of broad-spectrum antibiotics has a deleterious effect on the host microbiome [32]. These are the main reasons why the FDA has approved fewer broad-spectrum antibiotics for use in the last two decades. Narrow-spectrum antimicrobials that do not select for cross-resistance and that decrease the possible damage to the host microbiome are urgently needed in the fight against infections caused by multidrug-resistant pathogens [33].

Furthermore, the antibacterial activity of the selected extracts is expected to be enhanced by isolation of the active compounds from these complex mixtures of compounds. Moreover, synergistic or antagonistic interactions between the different components of the extracts are also possible. All of these factors must be taken into consideration in order to initiate the processes of isolation and identification of the corresponding antibacterial agent(s).

As far as we are aware, there are no previous studies of the antibacterial activity of *H. (Rhizoniera) curacaoensis* or *A. dilatata*. However, antibacterial activities of the remaining six sponges have been reported. Thus, agelasidines with antifungal activity against *Candida albicans* have been extracted from *A. citrina*, one of the most promising crude extracts in the present study, and (-)-agelasidine C (MIC 0.5 mg/L) was found to be the most active [34]. More recently, new pyrrole-imidazole alkaloids (denominated citrinamines) were isolated from the same species and were found to be able to inhibit the growth of the Gram-positive bacteria *Micrococcus luteus* and *Mycobacterium phlei* [35]. Inhibition of the growth of *E. coli* and *Pseudomonas putida* by extracts of this sponge has also been reported [36]. Furthermore, the presence of antibacterial activity in its extracts has also been reported in a study of the inhibition of growth of some non-pathogenic marine bacteria [37]. In the present study, we detected high activity of the extract from *A. citrina* (MIC ≤ 8 mg/L) against the four pathogens tested.

Antimicrobial data on the sponge *A. crassa* has been published previously under the synonymous names of *Ianthella basta* and *Ianthella ardis.* Bastadin 1-6, hemibastadins 1-3 and hemibastadinols 1-3 were isolated from *I. basta*, all displaying antibacterial activity against *Neisseria gonorrhoeae*, *Enterococcus faecalis*, and *S. aureus* [38,39]. Recently, dibromohemibastadin-1, which showed a potent inhibition of biofilm formation of *Paracoccus* sp. 4M6 and *P. aeruginosa* PAO1 (at 10 µM) and *quorum sensing* inhibition of *E. coli* pSB401 (at 8–16 µM), was isolated from the same species [40]. (-)-Aeroplysinin-1 and ianthelline were reported from *I. ardis*, both displaying antibacterial activity, the latter active against *S. aureus* [41,42].

Previous studies on *A. compressa* reported the isolation and identification of a new alkyl pyridine alkaloid, 8,8′-dienecyclostellettamine, which displayed a potent antibacterial activity against *E. coli*, *P. aeruginosa*, *Cryptococcus neoformans*, and methicillin-resistant *S. aureus* (MRSA), with IC_50_ values of 1.3, 2.1, 2.5, and 0.25 mg/L, respectively, as well as antifungal activity against *C. albicans* and *Aspergillus fumigatus* with IC_50_ values of 0.4 and 0.3 mg/L, respectively [43]. Extracts from the same species were also active against marine strains [44] and *E. faecalis* [45].

Batzelladine L was reported from the sponge *M. arbuscula* and displays antifungal activity against *Aspergillus flavus* [46]. Previously, *M. arbuscula* was reported under the name of *M. unguifera*, describing the isolation of ptilomycalin A, batzelladines L, M, C, dehydrobatzelladine C, crambescidine 800, and 16β-hidroxycrambescidin 359, which showed antibactertial activity against MRSA, *P. aeruginosa*, and *Mycobacterium intracellulare*, with MIC values ranging between 0.31 and 20.0 mg/L [47].

The antimicrobial agent sceptrin has been isolated from *A. sceptrum*, yielding MICs of 15 mg/L for pathogenic species such as *S. aureus*, *P. aeruginosa* and *C. albicans* [48,49]. The antibacterial activity displayed by the extract of *A. sceptrum* against *K. pneumoniae* and *P. aeruginosa* (MICs of 256 and 64 mg/L, respectively) in the present study is consistent with the aforementioned findings. Finally, the antibacterial activity of extracts of species belonging to the genus *Dysidea* has previously been reported; e.g., new sesquiterpenes isolated from the sponge *Dysidea* sp. displayed antibacterial activity against *E. coli* and *S. aureus* [50]. In the present study, *Dysidea* sp. extracts displayed antibacterial activity against *A. baumannii* and *S. aureus* (with MICs of 16 and 32 mg/L, respectively).

### 2.2. Antibacterial Activity and Bioassay-Guided Isolation of A. compressa Crude Extract

The crude extract of *A. compressa* was submitted to a bioassay-guided fractionation because of the promising antibacterial activity detected (MIC of 32 mg/L for the four bacterial species tested in the microdilution assay). A combination of solid phase extraction (SPE) with an RP-18 cartridge (Merck KGaA) and reverse phase high performance liquid chromatography (RP-HPLC) was used to fractionate the organic extract and purify the active compounds. The bioassay-guided fractionation of the organic extract of *A. compressa* allowed us to separate fractions R3 and R4, which were eluted by SPE with, respectively, 1:1 and 1:2 mixtures of H_2_O and CH_3_OH. Both fractions yielded the highest (and identical) MIC values (2–16 mg/L, Table 2 and Table 3) for all pathogens tested. Taking into account that the ^1^H NMR spectra of both fractions were very similar, suggesting that they contained similar compounds, and the fact that fraction R4 appeared to be purer than the R3 fraction, the former was chosen for further fractionation by RP-HPLC. The RP-HPLC chromatogram displayed a major peak (subfraction R4H2) with a retention time of 16.8 min (Appendix A). 

The ^1^H-NMR spectrum of subfraction R4H2 shows proton aromatic signals at *δ*_H_ 8.88 (s), 8.85 (d), 8.41 (d), and 8.02 (dd), suggesting the presence of pyridine rings, proton olefinic signals at *δ*_H_ 5.73 (m), 5.58 (m), and 4.60 (t), indicating the presence of double bonds, and proton signals at *δ*_H_ 3.68 (d), 2.20 (m), 2.02 (m), and 1.39 (m) ppm, characteristic of carbon aliphatic chains (Appendix A). 

The presence of a pyridine moiety and olefinic carbons was confirmed by the carbon resonances in the ^13^C-NMR spectrum, at *δ*_C_ 146.5, 145.2, 144.5, 143.5, 135.3, 129.0, and 125.3, which also displayed the presence of carbon aliphatic chains with carbon resonances at *δ*_C_ 63.0, 32.6, 31.1, 30.6, 30.6, 30.4, 30.2, 28.4, and 27.3 ppm (Appendix A). Comparison of the 1D NMR data with previously reported data allowed us to identify an inseparable mixture of amphitoxins and halitoxins in subfraction R4H2 (Figure 1) [51,52]. The structure of amphitoxin differs from that of halitoxin in that it has an additional carbon-carbon double bond in the alkyl chain that joins the pyridinium rings. Their similar structures make separation difficult and all attempts to separate this mixture were unsuccessful [46,53,54,55]. Subfraction R4H2 displayed higher antibacterial activity than fraction R4, yielding a MIC value of 2–4 mg/L for different strains of *A. baumannii*, *K. pneumoniae*, and *P. aeruginosa* and a MIC value of 1 mg/L for both strains of *S. aureus* (Table 2 and Table 3).

Sponges of the order Haplosclerida, to which *Amphimedon compressa* belongs, constitute well-known sources of alkylpyridine and alkylpiperidine alkaloids [40,56]. More specifically, halitoxins and amphitoxins are cyclic alkylpyridine oligomers of challenging separation and structural characterization due to their polycharged structures and high molecular weights [51,52,53]. They exert a multitude of biological actions. Regarding antibacterial activity, these compounds are able to inhibit bacterial attachment of *Vibrio harveyi*, a motile marine bacterium, but have not been reported to inhibit growth of the species [57]. Furthermore, these compounds have also been reported to show antibacterial activity against marine environmental isolates (MICs 4–256 mg/L) [55], as well as antifungal activity [47]. However, the antibacterial activity of halitoxins and amphitoxins detected in this study against the main multidrug-resistant human pathogens had not yet previously been observed.

### 2.3. Antibacterial Activity and Bioassay-Guided Isolation of A. citrina Crude Extract

The crude extract of *A. citrina* was submitted to a bioassay-guided fractionation because of the promising antibacterial activity detected (MIC of 8 mg/L against the Gram-negative and 0.5 mg/L against the Gram-positive bacteria species tested in the microdilution assay). 

The crude extract was partitioned using the modified Kupchan procedure [58] to afford a very active dichloromethane fraction (FD), with MICs of 2–32 mg/L (Table 4), which was submitted to solid phase extraction (SPE) using an RP-18 cartridge to give seven subfractions (R1–R7). RP-HPLC purification of R2, which appeared as the most pure fraction (based on its ^1^H NMR spectrum), with a MIC value of 4–8 mg/L against *S. aureus* (Table 5), led to the isolation of a pure compound with a retention time of 35.2 min (see RP-HPLC chromatogram in Appendix A). Its high resolution electrospray mass spectrometry (HRESIMS) showed the [M + H]^+^ ion adduct at *m/z* 422.3266, and the ^1^H and ^13^C NMR spectral data (Appendix A) and its optical rotation value matched with those of (-)-agelasine B (Figure 2), isolated from an unidentified sponge of the *Agelas* genus collected in the Okinawan sea [59,60].

(-)-Agelasine B displays higher antibacterial activity against the two Gram-positive bacteria *S. aureus* strains (2 mg/L) than fraction R2 (4 and 8 mg/L against ATCC 29213 and USA300 strains, respectively), as shown in Table 5. However, (-)-agelasine B did not show significant activity against the Gram-negative bacteria (*A. baumannii*, *K. pneumoniae*, and *P. aeruginosa*). The identification of the remaining compounds responsible of the antibacterial activity against those bacteria are on the way.

The antimicrobial activity (MIC) of (-)-agelasine B has been reported against *Saccharomyces cerevisiae* ATCC 188224 (10 mg/L) [61], *Staphylococcus epidermidis* 13889 (1.56 mg/L), *E. faecalis* 12964 (6.25 mg/L), *E. faecium* 12367 (3.13 mg/L), *S. aureus* 12732 (0.78 mg/L), MRSA (3.13 mg/L), *Candida albicans* IFO-1269 (>12.5 mg/L), and *Cryptococcus neoformans* TIMM-0354 (3.13 mg/L) [62], *Mycobacterium smegmatis* (3.13 mg/L) [63,64], *M. bovis* (6.25–12.5 mg/L) [64], and *Proteusbacillus vulgaris* (18.75 mg/L) [65]. However, (-)-agelasine B did not display antimicrobial activity against *P. aeruginosa* ATCC 15442, *S. aureus* ATCC 6538, *Asperigillus niger* (ATCC 9642) at 200 mg/L [62], and *E. coli* (>150 mg/L). Its enantiomer, (+)-agelasine B, the isolation of which was reported in 2017, shows a potent antibacterial activity against clinical MRSA and methicillin-susceptible *S. aureus* (MSSA) [66].

## 3. Material and Methods 

### 3.1. General Experimental Procedures 

Optical rotations were measured on a JASCO DIP-1000 polarimeter, with a Na (589 nm) lamp and filter. ^1^H, ^13^C, and 2D NMR spectra were recorded on a Bruker Avance 500 spectrometer at 500 and 125 MHz, respectively, using CD_3_OD and DMSO-d_6_. Low resolution electrospray mass spectrometry (LRESIMS) and high resolution electrospray mass spectrometry (HRESIMS) experiments were carried out on the Thermo LTQ Orbitrap Discovery system. HPLC separations were performed using an Agilent 1100 liquid chromatography system equipped with a solvent degasser, quaternary pump, and diode array detector (Agilent Technologies, Waldbronn, Germany) and a semipreparative reversed phase column Luna C18, 5 µm, 100 Å, 250 × 10 mm. Precoated silica gel plates (Merck, Kieselgel 60 F254, 0.25 mm) were used for TLC analysis, and the spots were visualized under a UV light (254 nm) or by heating the plate pretreated with H_2_SO4/H_2_O/AcOH (1:4:20).

### 3.2. Animal Collection and Identification

Samples of animals were collected by snorkelling and SCUBA diving in different coastal zones of the Yucatan Peninsula, Mexico, during three different periods: September–December 2016, January–March 2017, and September 2018. The selected species were collected from two different regions: Mexican Caribbean (Cozumel Island, Rio Indio, Mahahual, and Bermejo, Quintana Roo) and Campeche Bank (Alacranes Reef and Progreso, Yucatan) in areas with high biological diversity, such as coral reefs, islands, and mangrove forests (Figure 3).

The samples were labelled with a code according to the collection site, stored in plastic bags, and chilled on ice during transport to the laboratory. Voucher specimens of sponges were deposited in the Phylum Porifera Gerardo Green National Collection of the Institute of Marine Sciences and Limnology (ICMyL) at the National Autonomous University of Mexico (UNAM), Mexico City, while voucher specimens of ascidians were deposited in the Marine Biology Collection at the Autonomous University of Yucatan (UADY) in Yucatan, Mexico.

The sponges were identified at the ICMyL-UNAM (Mexico), while the ascidians were identified at the University of Vigo (Spain) and the Autonomous University of Yucatan (Mexico). Taxonomic information, collection sites, and previous antimicrobial activity records of the species or genus studied of the 64 selected marine organisms, are shown in Table 6.

### 3.3. Preparation of the Organic Extracts

Tissue slices from each species were exhaustively extracted with a mixture of dichloromethane-methanol (1:1), i.e., three times each with 500 mL of solvent (1.5 L total volume) at 25 °C for 24 h. The solvent was filtered and then removed under vacuum at 40 °C with a rotatory evaporator. The extracts were stored at −20 °C in tightly sealed glass vials.

### 3.4. Bioassay-Guided Isolation of the A. compressa Crude Extract

Whole bodies of *A. compressa* (wet weight, 509.2 g; dry weight, 64.0 g) were sliced and exhaustively extracted, as previously described, to yield 8.3 g of a crude residue. Fractionation of 8.2 g of the residue by solid phase extraction (SPE), using a stepped gradient from H_2_O to CH_3_OH and then CH_2_Cl_2_ (H_2_O (100%), H_2_O/CH_3_OH (2:1, 1:1, and 1:2), CH_3_OH (100%), CH_3_OH/CH_2_Cl_2_ (1:1), and CH_2_Cl_2_ 100%), yielded seven fractions (R1–R7). The fractions were concentrated under reduced pressure producing the following weights: R1: 6.7 g, R2: 213.3 mg; R3: 71.6 mg; R4: 74.7 mg; R5: 931.4 mg; R6: 192.3 mg; R7: 11.1 mg. In the evaluation of the fractions in the both microdilution assays (Table 2 and Table 3), fractions R3 and R4 displayed antibacterial activity. Part of the most active fraction R4 (50 mg) was further fractioned by HPLC, the mobile phase consisted of: (A) H_2_O with 0.04% of trifluroacetic acid; (B) CH_3_OH with 0.04% of trifluoracetic acid at a flow rate of 2.0 mL/min. A combination of gradient and isocratic elution was used, starting with 90% A and 10% B, changing to 67% of B in 10 min, followed by 5 min isocratic at 67% of B, 2 min gradient from 67% to 75% of B, 23 min of isocratic at 75% of B, and finally, changing to 100% of B in 5 min. A major peak, subfraction R4H2, eluted with a retention time of 16.8 min, was collected and concentrated under reduced pressure to produce 8.0 mg of yellow solid which was identified by NMR analysis as a mixture of alkylpyridine oligomers halitoxins and amphitoxins. Bioassay-guided fractionation was performed using an antibacterial activity assay against the four bacterial pathogens, as detailed in below. 

### 3.5. Bioassay-Guided Isolation of the A. citrina Crude Extract

Sliced bodies of *A. citrina* (wet weight, 729.6 g; dry weight, 375.3 g) were exhaustively extracted, as previously described, to give 6.1 g of a crude residue. Liquid-liquid fractionation of 6.0 g between H_2_O/CH_2_Cl_2_ (1:1 v/v) gave aqueous and an organic phases. The aqueous phase was extracted with n-butanol (200 mL) to yield 217.0 mg of the final aqueous fraction (WW) and 756.0 mg of the *n*-butanol fraction (WB), after removal the solvents under reduced pressure. The organic phase was concentrated under reduced pressure and was further partitioned between 10% aqueous CH_3_OH (400 mL) and hexane (2 × 400 mL) to give, after removing the solvent under reduced pressure, 672.2 mg of the hexane fraction (FH). The H_2_O content (% v/v) of the methanolic fraction was adjusted to 50% aqueous CH_3_OH, and the mixture was extracted with CH_2_Cl_2_ (100 mL) to afford, after removing the solvent under reduced pressure, 3.6 g of the CH_2_Cl_2_ fraction (FD) and 755.8 mg of the remaining aqueous methanolic fraction (FM). The CH_2_Cl_2_ and aqueous methanolic fractions displayed antibacterial activity in the microdilution assays (Table 4). Part of the CH_2_Cl_2_ fraction (3.6 g) was subjected to a solid phase extraction (SPE), using a stepped gradient from H_2_O to CH_3_OH and then CH_2_Cl_2_ (H_2_O (100%), H_2_O/CH_3_OH (2:1, 1:1, and 1:2), CH_3_OH (100%), CH_3_OH/CH_2_Cl_2_ (1:1), and CH_2_Cl_2_ 100%), to yield, after removing the solvents under reduced pressure, seven fractions: R1 (2.5 g); R2 (181.3 mg); R3 (191.3 mg); R4 (391.1 mg); R5 (315.9 mg); R6 (5.3 mg); and R7 (11.5 mg). All of them displayed antibacterial activity against *S. aureus* (Table 5). The active fraction R2 (181.3 mg) was further fractioned by HPLC. The mobile phase consisted of (A) H_2_O with 0.04% of trifluroacetic acid, and (B) CH_3_OH with 0.04% of trifluoracetic acid, and the analysis were run at a flow rate of 2.0 mL/min. A combination of gradient and isocratic elution was used, starting with 30% B, increasing to 100% of B in 30 min, followed by 10 min isocratic at 100% of B. A major peak, subfraction R2H13, eluted with a retention time of 35.2 min, was collected and concentrated under reduced pressure to produce 3.0 mg of white solid which was identified by NMR and MS analysis as (-)-agelasine B. Bioassay-guided fractionation was performed using an antibacterial activity assay against the four bacterial pathogens, as detailed in below. 

**(-)-agelasine B**. ${\left[\alpha \right]}_{D}^{25}$ −11.0 (*c* 0.17, MeOH); ^1^H and ^13^C NMR spectra see SM; (+)-HRESIMS *m/z* 422.3266 [M + H]^+^, (calcd. for C_26_H_40_N_5_, *m/z* 422.3284).

### 3.6. Antimicrobial Activity Assays

#### 3.6.1. Bacterial Strains and Culture Preparation

The bacterial strains used to study the antibacterial activity of the crude extracts (Appendix A) were the Gram-negative pathogens *A. baumannii* (strain ATCC 17978), *P. aeruginosa* (strain ATCC 27853), and *K. pneumoniae* (strain ATCC 700603), and the Gram-positive pathogen *S. aureus* (strains ATCC 29213). For bioassay-guided fractionation of *A. compressa*, the following bacterial strains were also used: *A. baumannii* ABRIM, *K. pneumoniae* Kp3380, *P. aeruginosa* PAO1, and *S. aureus* USA 300 LAC. Strain PAO1 is a reference strain, commonly used in research, and the other three strains are clinical isolates collected from samples from the Complexo Hospitalario Universitario A Coruña (CHUAC) hospital in Spain.

Gram-negative and Gram-positive strains were routinely grown or maintained in Luria-Bertani (LB), and in Trypticase soya broth (TSB) media, respectively, supplemented with 2% agar or the antibiotic ampicillin (30 mg/L), when needed. All strains were grown at 37 °C and stored in 10% glycerol at −80 °C. 

#### 3.6.2. Microdilution Method: Minimum Inhibitory Concentration

The minimum inhibitory concentrations (MICs) were determined by the broth microdilution method (CLSI, 2012). Briefly, the bacterial strains were cultured overnight at 37 °C in Mueller Hinton II agar plates (Becton Dickinson) and the turbidity of the bacterial suspensions was standardized at 0.5 on the McFarland scale to establish the inocula. The crude extracts of the test samples were dissolved in dimethylsulfoxide (DMSO). Two-fold serial dilutions of the extracts in Mueller Hinton II broth medium (Sigma) were carried out in 96-wells microplates, to produce a range of extract concentrations of 0.5–256 mg/L. DMSO was present at a maximum concentration of 2.5% v/v in the well containing the highest concentration of extract (256 mg/L). One well in each row contained growth media and bacterial suspension and was used as a positive growth control. Another well, containing medium only, was used as a negative control. Solvent controls of DMSO and growth medium were included to determine whether the concentrations used interfered with bacterial growth. The MIC was evaluated after incubation 20–24 h to 37 °C and was established as the lowest concentration of the compound at which the bacterial strains did not grow. All extracts were tested in triplicate.

MIC assays were also performed with ampicillin, cefoxitin, cefotaxime, ceftazidime, cefepime, imipenem, ciprofloxacin, and tobramycin against *A. baumannii*, *K. pneumoniae*, and *P. aeruginosa* strains, and with cefoxitin, oxacillin, erythromycin, clindamycin, and vancomycin against *S. aureus* strains (Appendix A).

## 4. Conclusions

There is an urgent need for successful treatments for patients with infections caused by multidrug-resistant bacteria. Developing new antibiotics that contribute to fighting against antibacterial resistance is one of the main current objectives in public health. 

A total of 64 marine organisms, 51 sponges (Porifera), and 13 ascidians (Chordata), selected on the basis of chemotaxonomic criteria, were collected on the coast of the Yucatan Peninsula in Mexico and organic extracts were obtained from each. One aliquot of each extract was submitted to in vitro antibacterial screening against four species of multidrug-resistant (MDR) bacterial pathogens: the Gram-negative *A. baumannii*, *K. pneumoniae*, and *P. aeruginosa*, and the Gram-positive *S. aureus*. MICs (determined by microdilution assay) indicated antibacterial activity of nine extracts from eight sponges: *A. citrina*, *A. dilatata*, *A. sceptrum*, *A. crassa* (collected from two different locations: Mahahual, Quintana Roo and Alacranes Reef), *A. compressa*, *Dysidea* sp., *M. arbuscula*, and *H. (Rhizoniera) curacaoensis*. Some of the extracts showed similar MIC values to known antibiotics, such as penicillins and aminoglycosides. Isolation of pure compounds from the complex mixtures constituting the extracts is expected to yield greater antibacterial activity. Preliminary studies on two of the active sponge extracts were performed using a bioassay-guided fractionation methodology. Thus, the active extract from the sponge *A. compressa* yielded an unseparable mixture of halitoxins and amphitoxins which displayed notable antibacterial in vitro activity against all four pathogenic bacteria. This is the first report of the antimicrobial activity of halitoxins and amphitoxins against major multidrug-resistant human pathogens. On the other hand, (-)-agelasine B was isolated from the sponge *A. citrina* as the major component responsible of the potent antibacterial activity of its extract against the Gram-positive *S. aureus*. The inactivity (-)-agelasine B against the three Gram-negative bacteria tested (*A. baumannii*, *K. pneumoniae*, and *P. aeruginosa*) was indicative of the presence of additional active compounds not isolated yet.

These organisms will be subjected to further detailed analysis to isolate biologically active molecules in the search for new compounds. Furthermore, the promising antibacterial activities detected in this study indicate the coast of Yucatan Peninsula as a potential source of a great variety of marine organisms worthy of further research. This type of study can serve as a basis for the development of new antibacterial agents effective against the principal multidrug-resistant bacterial pathogens for which the therapeutic options are increasingly scarce. This study constitutes the first report of antibacterial activity for a wide collection of sponges and ascidians collected on the coast of the Yucatan Peninsula.

## Figures and Tables

**Figure 1 marinedrugs-18-00369-f001:**
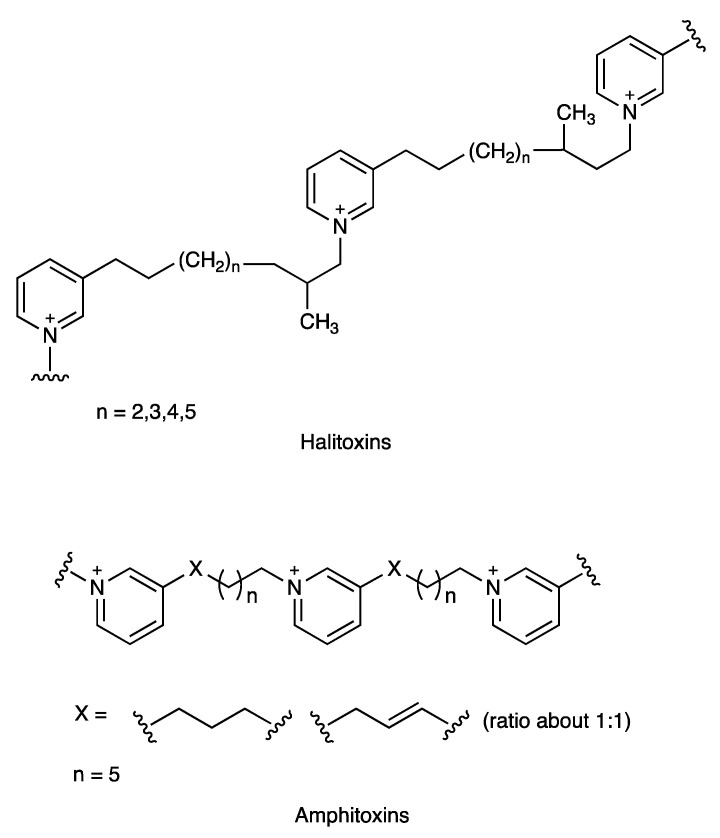
Substructure of the two closely related halitoxins and amphitoxins present as an inseparable mixture in the R4H2 subfraction from *A. compressa*.

**Figure 2 marinedrugs-18-00369-f002:**
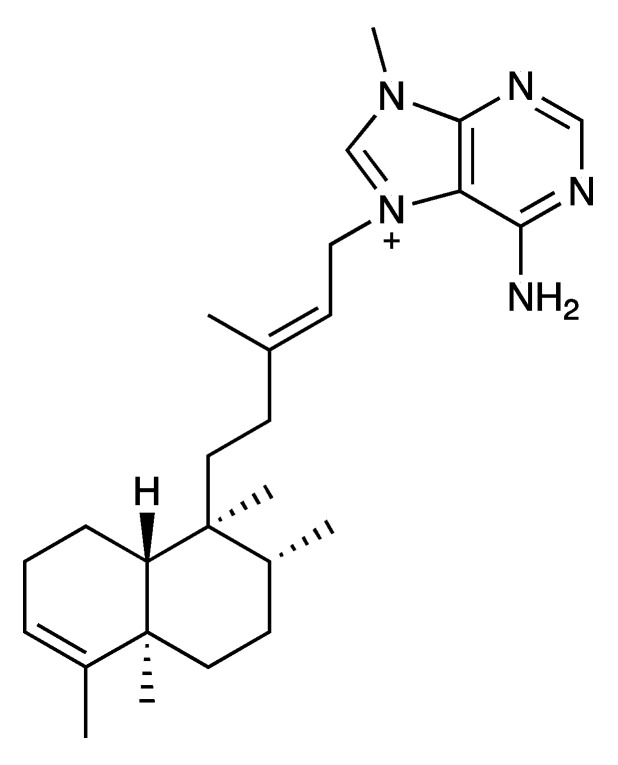
Structure of the (-)-agelasine B, isolated from *Agelas citrina*.

**Figure 3 marinedrugs-18-00369-f003:**
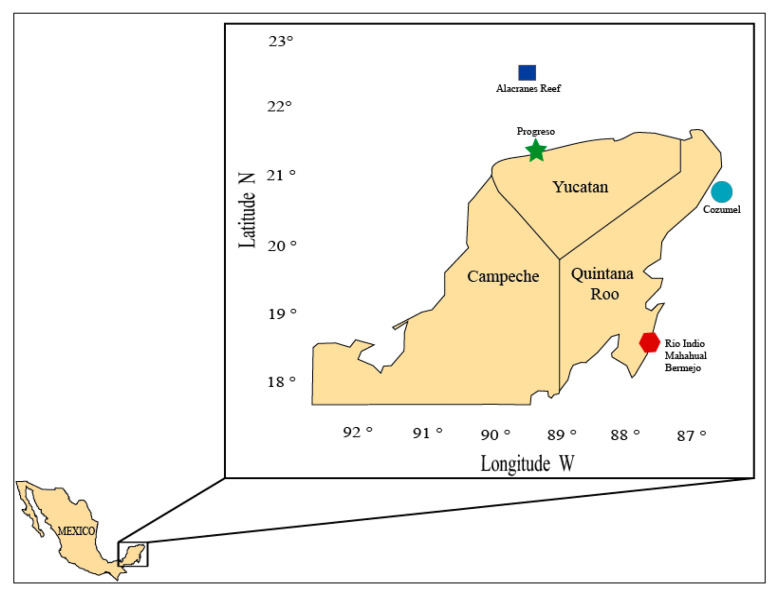
Sites of collection of marine organisms in the Yucatan Peninsula, Mexico.

**Table 1 marinedrugs-18-00369-t001:** Minimum inhibitory concentrations (MICs in mg/L) of organic extracts of marine organisms from the Yucatan Peninsula.

Phylum	Order	Family, Species (Code)	*A. baumanni* ATCC 17978	*K. pneumonia* ATCC 700603	*P. aeruginosa* ATCC 27823	*S. aureus* ATCC 29213
**Chordata**	Aplousobranchia	Clavelinidae ***Clavelina* sp.** (T18-M1)	≥512	≥512	≥512	≥512
		Didemnidae ***Didemnum perlucidum*** (E8-2)	≥512	≥512	≥512	>512
		***Didemnum* sp.** (T18-M4)	≥512	≥512	≥512	≥512
		***Didemnum* sp.** (E01)	≥512	≥512	≥512	≥512
		***Trididemnum solidum*** (E7-2)	≥512	≥512	>512	>512
		***Polysyncraton* sp.** (EY18-8)	≥512	≥512	≥512	≥512
		Polycitoridae ***Eudistoma amanitum*** (RIO18-T1)	≥512	≥512	≥512	≥512
		***Eudistoma* sp.** (TY18-2)	≥512	≥512	≥512	≥512
		Polyclinidae ***Polyclinum* sp.** (T18-M5)	≥512	≥512	≥512	≥512
	Phlebobranchia	Ascidiidae ***Phallusia nigra*** (TY18-1)	≥512	≥512	≥512	≥512
		Perophoridae ***Ecteinascidia* sp.** (T18-M2)	≥512	≥512	≥512	≥512
	Stolidobranchia	Molgulidae ***Molgula* sp.** (T18-M6)	≥512	≥512	≥512	≥512
		Styelidae ***Polycarpa* sp.** (E41)	≥512	≥512	≥512	≥512
**Porifera**	Agelasida	Agelasidae ***Agelas citrina*** (CZE56)	**8**	**8**	**8**	**0.5**
		***Agelas clathrodes*** (E27-2)	≥512	≥512	≥512	>512
		***Agelas clathrodes*** (MA18-10)	≥512	≥512	≥512	≥512
		***Agelas dilatata*** (E25-1)	128	**64**	**32**	**64**
		***Agelas sceptrum*** (E26-2)	≥512	256	**64**	>512
	Axinellida	Heteroxyidae ***Myrmekioderma gyroderma*** (CZE18)	≥512	≥512	≥512	≥512
		Raspailiidae ***Ectyoplasia ferox*** (MA18-9)	≥512	≥512	≥512	≥512
		***Ectyoplasia* sp.** (MA18-13)	≥512	≥512	≥512	≥512
	Chondrillida	Chondrillidae ***Chondrilla caribensis f. hermatypica*** (MA18-6)	≥512	≥512	≥512	≥512
		***Chondrilla* sp.** (RIO18-1)	≥512	≥512	≥512	≥512
	Clathrinida	Clathrinidae ***Clathrina* sp.** (EY18-10)	≥512	≥512	≥512	≥512
		Leucettidae ***Leucetta floridana*** (E2-2)	128	256	≥512	128
	Clionaida	Clionaidae ***Cliona delitrix*** (EY18-1)	≥512	≥512	≥512	≥512
		***Cliona varians*** (EY18-3)	≥512	≥512	≥512	≥512
	Dictyoceratida	Dysideidae ***Dysidea* sp.** (EY18-12)	**16**	≥512	≥512	**32**
		Irciniidae ***Ircinia felix*** (E9-2)	≥512	≥512	≥512	>512
		***Ircinia felix*** (MA18-11)	≥512	≥512	≥512	≥512
		***Ircinia strobilina*** (E24-2)	≥512	≥512	≥512	>512
		***Ircinia strobilina*** (E52)	≥512	≥512	≥512	≥512
		Spongiidae ***Spongia tubulifera*** (E11-2)	≥512	≥512	≥512	≥512
	Haplosclerida	Callyspongiidae ***Callyspongia longissima*** (E28)	≥512	≥512	≥512	≥512
		***Callyspongia plicifera*** (E31)	≥512	≥512	≥512	≥512
		***Callyspongia vaginalis*** (E16)	≥512	≥512	≥512	≥512
		Chalinidae ***Haliclona (Rhizoniera) curacaoensis*** (EY18-4)	**4**	**16**	**32**	**4**
		Niphatidae ***Amphimedon compressa*** (E29)	**32**	**32**	**32**	**32**
		***Niphates digitalis*** (E15)	≥512	≥512	≥512	≥512
		***Niphates erecta*** (E49)	≥512	≥512	≥512	≥512
		***Niphates erecta*** (MA18-7)	≥512	≥512	≥512	≥512
		***Niphates erecta*** (MA18-12)	≥512	≥512	≥512	≥512
		Petrosiidae ***Xestospongia muta*** (EP)	≥512	≥512	≥512	≥512
	Homoscleropho-rida	Plakinidae ***Plakinastrella onkodes*** (E3)	≥512	≥512	≥512	≥512
	Poecilosclerida	Crambeidae ***Monanchora arbuscula*** (E35)	≥512	≥512	≥512	**16**
		Microcionidae ***Clathria gomezae*** (EY18-11)	≥512	≥512	≥512	≥512
		***Clathria virgultosa*** (E7-E34)	≥512	≥512	≥512	≥512
		Mycalidae ***Mycale laevis*** (MA18-1)	≥512	≥512	≥512	≥512
		***Mycale laevis*** (MA18-5)	≥512	≥512	≥512	≥512
	Scopalinida	Scopalinidae ***Scopalina ruetzleri*** (MA18-5)	≥512	≥512	≥512	≥512
		***Scopalina ruetzleri*** (E53)	≥512	≥512	≥512	≥512
		***Scopalina ruetzleri*** (EY18-7)	≥512	≥512	≥512	≥512
	Suberitida	Halichondriidae ***Halichondria melanadocia*** (E18-M1)	≥512	≥512	≥512	≥512
		Suberitidae ***Aaptos* sp.** (E38)	≥512	≥512	≥512	≥512
	Tethyida	Tethyidae ***Tethya* sp.** (E20)	≥512	≥512	≥512	≥512
	Tetractinellida	Geodiidae ***Melophlus hajdui*** (E4)	≥512	≥512	≥512	≥512
		Tetillidae ***Cinachyrella kuekenthali*** (MA18-2)	≥512	≥512	≥512	≥512
	Verongiida	Aplysinidae ***Aiolochroia crassa*** (E50)	**64**	128	>128	**32**
		***Aiolochroia crassa*** (MA18-4)	**32**	128	128	**64**
		***Aplysina cauliformis*** (E36)	≥512	≥512	≥512	≥512
		***Aplysina fistularis*** (E46)	≥512	≥512	≥512	≥512
		***Aplysina fulva*** (E42)	≥512	≥512	≥512	≥512
		***Aplysina fulva*** (EY18-5)	≥512	≥512	≥512	≥512
		***Aplysina muricyanna*** (E47)	≥512	≥512	≥512	≥512
**Imipenem positive control** **Vancomycin positive control**			0.5	0.25	2	nt
			nt	nt	nt	1

nt: not tested. Bold numbers: high or intermediate MIC activities.

**Table 2 marinedrugs-18-00369-t002:** MICs (mg/L) of the SPE fractions separated from the crude extract of *A. compressa*.

Fraction	*A. baumannii* ATCC 17978	*K. pneumoniae* ATCC 700603	*P. aeruginosa* ATCC 27823	*S. aureus* ATCC 29213
R1	128	256	64	nt *
R2	128	256	256	Nt
R3	8	16	16	2
R4	8	16	16	2
R5	>512	>512	>512	Nt
R6	>512	>512	>512	Nt
R7	>512	>512	>512	Nt

* nt: no tested.

**Table 3 marinedrugs-18-00369-t003:** MICs (mg/L) of the most active SPE fractions and HPLC subfraction R4H2 from *A. compressa*.

Bacterial Strain		R3	R4	R4H2
*A. baumannii*	ATCC 17978	8	8	4
	ABRIM	8	8	4
*K. pneumoniae*	ATCC 700603	16	16	4
	Kp3380	8	8	2
*P. aeruginosa*	ATCC 27853	16	16	2
	PAO1	16	16	4
*S. aureus*	ATCC 29213	2	2	1
	USA 300	1	1	1

**Table 4 marinedrugs-18-00369-t004:** MICs (mg/L) of the liquid-liquid extraction fractions separated from the crude extract of *A. citrina*.

Fraction	*A. baumannii* ATCC 17978	*K. pneumoniae* ATCC 700603	*P. aeruginosa* ATCC 27823	*S. aureus* ATCC 29213
WW	>128	>128	>128	>128
WB	>128	>128	>128	>128
FH	>128	>128	>128	>128
FM	64	64	32	4
FD	16	32	32	2

**Table 5 marinedrugs-18-00369-t005:** MICs (mg/L) of the SPE fractions separated from the FD fraction of *A. citrina* and (-)-agelasine B isolated from subfraction R2.

Bacterial Strain		R1	R2	R3	R4	R5	R6	R7	(-)-agelasine B
*A. baumannii*	ATCC 17978	>64	64	16	>64	16	>64	32	>128
	ABRIM	>64	64	16	>64	16	>64	32	>128
*K. pneumoniae*	ATCC 700603	>64	>64	32	>64	>64	>64	32	>128
	Kp3380	64	64	8	64	16	>64	16	>128
*P. aeruginosa*	ATCC 27853	>64	64	16	>64	>64	>64	32	>128
	PAO1	>64	64	16	>64	>64	>64	64	>128
*S. aureus*	ATCC 29213	8	4	0.5	8	2	8	2	2
	USA 300	16	8	1	8	4	16	4	2

**Table 6 marinedrugs-18-00369-t006:** Taxonomic information, voucher numbers, site of collection, and previously reported antibacterial activity for the species under study, including synonymised names for the eight species that showed active.

Family	Species, (Code)	Site	Reported Antibacterial Activity	References
**Phylum:** Chordata**Order:** Aplousobranchia
Clavelinidae	*Clavelina* sp.(T18-M1)	Progreso, Yucatan (Mangrove)	Example of species of this genus: *C. pictus*. Stereoisomers of piclavins A2 to A4 displayed low activity against Gram-positive bacteria (*S.aureus*, *B. cereus* and *C. michiganensis*).	[67]
Didemnidae	*Didemnum perlucidum*(E8-2)	Rio Indio, Quintana Roo	Low activity against *S. aureus*, and not active against *E. coli* and *P*. *aeruginosa.*	[68]
*Didemnum* sp. (T18-M4)	Progreso, Yucatan (Mangrove)	Example of species of this genus:Antimicrobial activity of an unidentified *Didemnum* species against *E. faecalis*, *S. aureus*, *S. typhimurium*, *S. marcescens* and *P. aeruginosa*.	[69]
*Didemnum* sp. (E01)	Bermejo, Quintana Roo
*Trididemnum solidum* (E7-2)	Rio Indio, Quintana Roo	No previous reports for this species.	
*Polysyncraton* sp. (EY18-8)	Progreso, Yucatan	No previous reports for this genus.	
Polycitoridae	*Eudistoma amanitum*(RIO18-T1)	Río Indio, Quintana Roo	No previous reports for this genus.	
*Eudistoma* sp. (TY18-2)	Progreso, Yucatan	No previous reports for this genus.	
Polyclinidae	*Polyclinum* sp. (T18-M5)	Progreso, Yucatan (Mangrove)	The extract of a *Polyclinum* sp. yielded MICs of > 1000 mg/L against *S. aureus*, *E. coli* and *P. aeruginosa.*	[68]
**Order:** Phlebobranchia
Ascidiidae	*Phallusia nigra*(TY18-1)	Progreso, Yucatan	Low antimicrobial activity against *B. subtilis*, *S. aureus*, *E. aerogenes*, *E. coli*, *K. pneumoniae*, *P. aeruginosa*, *S*. *paratyphii*, *S. typhii* and *V. cholera.*	[70]
Perophoridae	*Ecteinascidia* sp. (T18-M2)	Progreso, Yucatan (Mangrove)	No previous reports for this genus.	
**Order:** Stolidobranchia
Molgulidae	*Molgula* sp. (T18-M6)	Progreso, Yucatan (Mangrove)	No previous reports for this genus.	
Styelidae	*Polycarpa* sp. (E41)	Alacranes Reef, Yucatan	No previous reports for this genus.	
**Phylum:** Porifera**Order:** Agelasida
Agelasidae	*Agelas citrina* **(CZE56)**	Cozumel, Quintana Roo	Antimicrobial activity against *E. coli* and inhibition of its *quorum sensing*. No antimicrobial activity against *C. violaceum*. Inhibition of *quorum sensing* at high concentrations.	[37]
*Agelas clathrodes*(E27-2)	Cozumel, Quintana Roo	Clathrodin did not display antimicrobial activity against *E. faecalis*, *S. aureus* and *E. col,* but displayed low antifungal activity against *C. albicans.*	[71]
*Agelas clathrodes*(MA18-10)	Mahahual, Quintana Roo
*Agelas dilatate* **(E25-1)**	Cozumel, Quintana Roo	No previous reports for this species.	
*Agelas sceptrum* **(E26-2)**	Cozumel, Quintana Roo	Sceptrin displayed antimicrobial activity against *S.aureus*, *B. subtilis* and *P. aeruginosa*.	[49,72]
**Order:** Axinellida
Heteroxyidae	*Myrmekioderma gyroderma*(CZE18)	Cozumel, Quintana Roo	No previous reports for this species.	
Raspailiidae	*Ectyoplasia ferox*(MA18-9)	Mahahual, Quintana Roo	No previous reports for this species.	
*Ectyoplasia* sp. (MA18-13)	Mahahual, Quintana Roo	No previous reports for this genus.	
**Order:** Chondrillida
Chondrillidae	*Chondrilla caribensis f. hermatypica* (MA18-6)	Mahahual, Quintana Roo	No inhibition of *S. aureus*, *S. epidermidis* or *E. coli* growth.	[73]
*Chondrilla* sp. (RIO18-1)	Río Indio, Quintana Roo	No previous reports for this genus.	
**Order:** Clathrinida
Clathrinidae	*Clathrina* sp. (EY18-10)	Progreso, Yucatan	No previous reports for this genus.	
Leucettidae	*Leucetta floridana*(E2-2)	Bermejo, Quintana Roo	No previous reports for this species.	
**Order:** Clionaida
Clionaidae	*Cliona delitrix*(EY18-1)	Progreso, Yucatan	*Quorum sensing* inhibition in *E.coli.*	[37]
*Cliona varians*(EY18-3)	Progreso, Yucatan	No observed antibacterial activity against *E. coli* or *C. violaceum*.	[37]
**Order:** Dictyoceratida
Dysideidae	*Dysidea* sp. **(EY18-12)**	Progreso, Yucatan		
Irciniidae	*Ircinia felix*(E9-2)	Rio Indio, Quintana Roo	*Quorum sensing* inhibition in *C. violaceum.* No effects on bacterial growth observed for this species. Antibacterial activity against *B. subtilis.*	[37,74]
*Ircinia felix*(MA18-11)	Mahahual, Quintana Roo
*Ircinia strobilina*(E24-2)	Cozumel, Quintana Roo	Antibacterial activity against *B. subtilis.* No inhibition of *E. coli* growth.	[74]
*Ircinia strobilina*(E52)	Bermejo, Quintana Roo
Spongiidae	*Spongia tubulifera*(E11-2)	Rio Indio, Quintana Roo	No previous reports for this species.	
**Order:** Haplosclerida
Callyspongiidae	*Callyspongia longissima*(E28)	Alacranes Reef, Yucatan	No previous reports for this species.	
*Callyspongia plicifera*(E31)	Alacranes Reef, Yucatan	Antibacterial activity against *E. coli.*	[74]
*Callyspongia vaginalis*(E16)	Cozumel, Quintana Roo	Antibacterial activity against *B. subtilis*	[74]
Chalinidae	*Haliclona (Rhizoniera)* *curacaoensis* **(EY18-4)**	Progreso, Yucatan	No previous reports for this species.	
Niphatidae	*Amphimedon compressa* **(E29)**	Alacranes Reef, Yucatan	Antibacterial activity of extracts against marine bacteria strains, *E. faecalis*, *P. aeruginosa* and *E. coli.*8,8′-dienecyclostellettamine showed a potent antibacterial activity against *E. coli*, *P. aeruginosa* and MRSA with IC_50_ values of 1.3, 2.1, 0.25 mg/L respectively.	[43,44,45,46]
*Niphates digitalis*(E15)	Cozumel, Quintana Roo	No previous reports for this species.	
*Niphates erecta*(E49)	Alacranes Reef, Yucatan	No previous reports for this species.	
*Niphates erecta*(MA18-7)	Mahahual, Quintana Roo
*Niphates erecta*(MA18-12)	Mahahual, Quintana Roo
Petrosiidae	*Xestospongia muta*(EP)	Alacranes Reef, Yucatan	No growth or *quorum sensing* inhibition of *E. coli* or *C. violaceum.*	[37]
**Order:** Homosclerophorida
Plakinidae	*Plakinastrella onkodes*(E3)	Bermejo, Quintana Roo	No previous reports for this species.	
**Order:** Poecilosclerida
Crambeidae	*Monanchora arbuscula***(E35)**Synonymised names:*M. unguifera*	Alacranes Reef, Yucatan	Ptilomycalin A, batzelladines L, M, C, dehydrobatzelladine C, crambescidine 800 and 16β-hidroxycrambescidin 359 were isolated of this species and showed MIC of between 0.31–20.0 mg/L against *S. aureus*, methicillin-resistant *S. aureus* (MRSA), *P. aeruginosa* and *M. intracellulare*.	[47]
Microcionidae	*Clathria gomezae*(EY18-11)	Progreso, Yucatan	No previous reports for this species.	
	*Clathria virgultosa*(E7-E34)	Alacranes Reef, Yucatan	No previous reports for this species.	
Mycalidae	*Mycale laevis*(MA18-1)	Mahahual, Quintana Roo	No previous reports for this species.	
*Mycale laevis*(MA18-5)	Mahahual, Quintana Roo
**Order:** Scopalinida
Scopalinidae	*Scopalina ruetzleri*(DNY)	Rio Indio, Quintana Roo	No previous reports for this species.	
*Scopalina ruetzleri*(E53)	Cozumel, Quintana Roo
*Scopalina ruetzleri*(EY18-7)	Progreso, Yucatan
**Order:** Suberitida
Halichondriidae	*Halichondria melanadocia*(E18-M1)	Progreso, Yucatan (Mangrove)	No previous reports for this species.	
Suberitidae	*Aaptos* sp. (E38)	Alacranes Reef, Yucatan	No previous reports for this species.	
**Order:** Tethyida
Tethyidae	*Tethya* sp. (E20)	Cozumel, Quintana Roo	No previous reports for this species.	
**Order:** Tetractinellida
Geodiidae	*Melophlus hajdui*(E4)	Bermejo, Quintana Roo	Antibacterial activity against *Mycobacterium* sp.	[75]
Tetillidae	*Cinachyrella kuekenthali* (MA18-2)	Mahahual, Quintana Roo	Antibacterial activity of aqueous and ethanolic extracts against different species of the genus *Staphylococcus* sp.	[76]
**Order:** Verongiida
Aplysinidae	*Aiolochroia crassa***(E50)**Synonymised names:*Pseudoceratina crassa, Ianthella basta* and *Ianthella ardis*	Alacranes Reef, Yucatan	Antibacterial activity of extracts against marine bacteria strains and *B. subtilis* with MIC of 0.4 mg/L.Bastadin 1-6, hemibastadins 1-3 and hemibastadinols 1-3 showed antibacterial activity againts *Neisseria gonorrhoeae*, *E. faecalis* and *S. aureus.*Dibromohemibastadin-1 showed potent antibacterial activity against the biofilm formation of *Paracoccus sp.* 4M6 and *P. aeruginosa* PAO1 (10 µM) and *quorum sensing* inhibition of *E. coli* pSB401 (8–16 µM) assays.(-)-Aeroplysinin-1 showed antibacterial activity.Ianthelline showed antibacterial activity against *S. aureus.*	[38,39,40,41,42,44,74]
*Aiolochroia crassa* **(MA18-4)**	Mahahual, Quintana Roo
*Aplysina cauliformis*(E36)	Alacranes Reef, Yucatan	Antibacterial activity against *M. tuberculosis* H37Rv.	[77]
*Aplysina fistularis*(E46)	Alacranes Reef, Yucatan	No previous reports for this species.	
*Aplysina fulva*(E42)	Alacranes Reef, Yucatan	Antibacterial activity against marine bacteria strains.	[44]
*Aplysina fulva*(EY18-5)	Progreso, Yucatan
*Aplysina muricyanna*(E47)	Alacranes Reef, Yucatan	No previous reports for this species.

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
