# Peer review of "Marine Organisms from the Yucatan Peninsula (Mexico) as a Potential Natural Source of Antibacterial Compounds"

_marinedrugs, 2020, doi:10.3390/md18070369_

Round 1
Reviewer 1 Report
Briefly, this manuscript describes the extraction and antibacterial screening of 51 sponges and 13 ascidians, collected on the coast of the Yucatan Peninsula in Mexico. The extracts were screened against 4 bacterial strains using broth microdilution assay. Extracts of 8 of the species, all sponges, displayed activity against some of the pathogens tested. From the sponge Amphimedon compressa, a mixture of known halitoxins and amphitoxins were purified and characterized. From Agelas citrina, (-)-agelasine B was isolated and profiled for activities. The results indicate that the variety of species found the coast of Yucatan Peninsula is a potential source for discovering novel antibacterial drug leads.
I find this work of good quality, whose experimental design is sound. The data is in general well-presented and discussed, however, there is scope for improvement in certain portions of the paper.
For practical reasons, I will list my comments in a chronological order. Some comments are very minor (grammatical faults) while others are more major. For the latter, I will add (MAJOR) to highlight their importance.
- R&D (L137): “K. pneumonia” should be changed to “Klebsiella pneumonia”.
- R&D (L140-142): I suggest changing “We have included in this study two strains of every bacterial species, displaying one of them a susceptible background to classical antimicrobials in therapy and showing the other one a multiresistant background (see Tables S1 and S2 in SM)” to “We have included in this study two strains of every bacterial species; one strain susceptible to classical antimicrobials and one a multi-resistant strain (see Tables S1 and S2 in SM)”.
- R&D (L146): I suggest deleting “species in”.
- R&D - Table 1: MIC-values of positive controls, are they also in mg/mL? If they are in μg/mL, please add this info.
- R&D (L225-232): I suggest to change the text to: “Antimicrobial data on the sponge A. crassa has been published previously under the synonymous names of Ianthella basta and Ianthella ardis. From I. basta were the compounds bastadin 1-6, hemibastadins 1-3 and hemibastadinols 1-3 isolated, all displaying antibacterial activity against Neisseria gonorrhoeae, Enterococcus faecalis and S. aureus [39,40]. Recently, dibromohemibastadin-1, which showed a potent inhibition of biofilm formation of Paracoccus sp. 4M6 and P. aeruginosa PAO1 (at 10 μM) and quorum sensing inhibition of E. coli pSB401 (at 8-16 μM), was isolated from the same species [41]. (-)-Aeroplysinin-1 and ianthelline were reported from I. ardis, both displaying antibacterial activity, the last one active against S. aureus [42,43].”
- R&D (L233-239): I suggest to change the text to: “Previous studies on A. compressa reported the isolation and identification of new alkyl pyridine alkaloids, which displayed potent antibacterial activities against E. coli, P. aeruginosa, Cryptococcus neoformans and methicillin-resistant S. aureus (MRSA) as well as antifungal activity against C. albicans and Aspergillus fumigatus [44]. Extracts from the same species were also active against marine strains and E. faecalis [44]. The compound 8,8´-dienecyclostellettamine, isolated of the A. compressa, showed potent antibacterial activity against E. coli, P. aeruginosa and MRSA with IC50 values of 1.3, 2.1, 0.25 μg/mL, respectively [45,46].
- R&D (L244-245): I suggest changing “methicillin-resistant S. aureus (MRS), P. aeruginosa and M. intracellulare with MIC values between” to “MRSA, P. aeruginosa and Mycobacterium intracellulare with MIC values ranging between”.
- R&D - Table 3: Data for A. baumannii should be aligned in the same row.
- R&D (L310-319): I suggest to change the text to: “The crude extract was partitioned using the modified Kupchan procedure [59] to afford a very active dichloromethane fraction (FD), with MICs of 2-32 mg/L (Table 4), which was submitted to Solid Phase Extraction (SPE) using an RP-18 cartridge to give seven subfractions (R1-R7). RP-HPLC purification of R2 which appeared as the most pure fraction (based on its 1H NMR spectrum), with a MIC value of 4-8 mg/L against S. aureus (Table 5), led to the isolation of a pure compound with a retention time of 35.2 min (see RP-HPLC chromatogram in Figure S4 in the SM). Its HRESIMS spectrum, showing the [M + H]+ ion adduct at m/z 422.3266, the 1H and 13C NMR spectral data (see Figures S5-7 in the SM), and its optical rotation value matched with those of (-)-agelasine B, isolated from an unidentified sponge of the Agelas genus collected in the Okinawan sea. [60,61]”.
- R&D (L334-339): I suggest to change the text to: “The antimicrobial activity (MIC) of (-)-agelasine B has been reported against Saccharomyces cerevisiae ATCC 188224 (10 μg/mL) [62], Staphylococcus epidermidis 13889 (1.56 μg/mL), E. faecalis 12964 (6.25 μg/mL), E. faecium 12367 (3.13 μg/mL), S. aureus 12732 (0.78 μg/mL), MRSA (3.13 μg/mL), Candida albicans IFO-1269 (>12.5 μg/mL) and Cryptococcus neoformans TIMM-0354 (3.13 μg/mL) [63], Mycobacterium smegmatis (3.13 μg/mL) [64,65], M. bovis (6.25-12.5 μg/mL) [64], and Proteusbacillus vulgaris (18.75 μg/mL) [66].”
- R&D (MAJOR): The authors should check their own MIC-data throughout the R&D section. The MIC-data are sometimes listed as mg/L and sometimes listed as mg/mL, both in tables and text. For instance, in R&D, Line 311 MICs are listed in mg/L and when referring to Table 4, the same values are presents as mg/mL! That is a thousand-fold difference. In addition, when citing literature they use μg/mL for comparison of data. I suggest that they use a common unit of measurement (like μg/mL) throughout the paper.
- M&M (L403-404): The authors write: “Liquid-liquid fractionation of 6.0 g between CH3OH/CH2Cl2 (1:1 v/v) gave aqueous (WW) and an organic fraction. The WW fraction was extracted with n-butanol (200 mL) to yield the n-butanol fraction (WB = 756.0 mg).” How do you obtain an aqueous fraction from the CH3OH/CH2Cl2 mixture? Should it be H2O/CH2Cl2 (1:1 v/v)? Please clarify, and also add yield of the WW fraction.
- M&M (L407): Add space between 400 and mL.
- M&M (L417-421): I suggest to change the text to: “The active fraction R2 (181.3 mg) was further fractioned by HPLC. The mobile phase consisted of (A) H2O with 0.04% of trifluroacetic acid, and (B) CH3OH with 0.04% of trifluoracetic acid, and the analysis were run at a flow rate of 2.0 mL/min. A combination of gradient and isocratic elution was used, starting with 30% B, increasing to 100% of B in 30 min, followed by 10 min isocratic at 100% of B.”.
- M&M (L469): I suggest changing “MICs” to “MIC”.
- M&M (L470): I suggest deleting «the”.
- Conclusion (L485): Use abbreviated names for Amphimedon compressa and Haliclona.
- Conclusion (L488): Change “estudies” to “studies”.
Author Response
Reviewer 1
Briefly, this manuscript describes the extraction and antibacterial screening of 51 sponges and 13 ascidians, collected on the coast of the Yucatan Peninsula in Mexico. The extracts were screened against 4 bacterial strains using broth microdilution assay. Extracts of 8 of the species, all sponges, displayed activity against some of the pathogens tested. From the sponge Amphimedon compressa, a mixture of known halitoxins and amphitoxins were purified and characterized. From Agelas citrina, (-)-agelasine B was isolated and profiled for activities. The results indicate that the variety of species found the coast of Yucatan Peninsula is a potential source for discovering novel antibacterial drug leads.
I find this work of good quality, whose experimental design is sound. The data is in general well-presented and discussed, however, there is scope for improvement in certain portions of the paper.
For practical reasons, I will list my comments in a chronological order. Some comments are very minor (grammatical faults) while others are more major. For the latter, I will add (MAJOR) to highlight their importance.
Comments to the Author
- R&D (L137): “K. pneumonia” should be changed to “Klebsiella pneumonia”.
Answer to reviewer 1: This was corrected in the revised manuscript following the reviewer’s comment.
Comments to the Author
- R&D (L140-142): I suggest changing “We have included in this study two strains of every bacterial species, displaying one of them a susceptible background to classical antimicrobials in therapy and showing the other one a multiresistant background (see Tables S1 and S2 in SM)” to “We have included in this study two strains of every bacterial species; one strain susceptible to classical antimicrobials and one a multi-resistant strain (see Tables S1 and S2 in SM)”.
Answer to reviewer 1: This was corrected in the revised manuscript following the reviewer’s comment.
Comments to the Author
- R&D (L146): I suggest deleting “species in”.
Answer to reviewer 1: This was corrected in the revised manuscript following the reviewer’s comment.
Comments to the Author
- R&D - Table 1: MIC-values of positive controls, are they also in mg/mL? If they are in μg/mL, please add this info.
Answer to reviewer 1: All MICs carried out in the study were quantified in mg/L and this information was added to Table 1 as it was suggested by the reviewer.
Comments to the Author
- R&D (L225-232): I suggest to change the text to: “Antimicrobial data on the sponge A. crassa has been published previously under the synonymous names of Ianthella basta and Ianthella ardis. From I. basta were the compounds bastadin 1-6, hemibastadins 1-3 and hemibastadinols 1-3 isolated, all displaying antibacterial activity against Neisseria gonorrhoeae, Enterococcus faecalis and S. aureus [39,40]. Recently, dibromohemibastadin-1, which showed a potent inhibition of biofilm formation of Paracoccus sp. 4M6 and P. aeruginosa PAO1 (at 10 μM) and quorum sensing inhibition of E. coli pSB401 (at 8-16 μM), was isolated from the same species [41]. (-)-Aeroplysinin-1 and ianthelline were reported from I. ardis, both displaying antibacterial activity, the last one active against S. aureus [42,43].”
Answer to reviewer 1: This was corrected in the revised manuscript following the reviewer’s comment. We have done a little change in the following sentence: Instead of
“From I. basta were the compounds bastadin 1-6, hemibastadins 1-3 and hemibastadinols 1-3 isolated,…”
We write
“Bastadin 1-6, hemibastadins 1-3 and hemibastadinols 1-3 were isolated from I. basta, ..”
Comments to the Author
- R&D (L233-239): I suggest to change the text to: “Previous studies on A. compressa reported the isolation and identification of new alkyl pyridine alkaloids, which displayed potent antibacterial activities against E. coli, P. aeruginosa, Cryptococcus neoformans and methicillin-resistant S. aureus (MRSA) as well as antifungal activity against C. albicans and Aspergillus fumigatus [44]. Extracts from the same species were also active against marine strains and E. faecalis [44]. The compound 8,8´-dienecyclostellettamine, isolated of the A. compressa, showed potent antibacterial activity against E. coli, P. aeruginosa and MRSA with IC50 values of 1.3, 2.1, 0.25 μg/mL, respectively [45,46].
Answer to reviewer 1: This was corrected in the revised manuscript following the reviewer’s comment. Additionally, we have combined some sentences that were repeated and correcting some references were corrected. The new paragraph is:
“Previous studies on A. compressa reported the isolation and identification of a new alkyl pyridine alkaloid, 8,8´-dienecyclostellettamine, which displayed a potent antibacterial activity against E. coli, P. aeruginosa, Cryptococcus neoformans and methicillin-resistant S. aureus (MRSA) with IC50 values of 1.3, 2.1, and 0.25 mg/L, respectively, as well as antifungal activity against C. albicans and Aspergillus fumigatus with IC50 values of 0.4 and 0.3 mg/L, respectively [44]. Extracts from the same species were also active against marine strains [45] and E. faecalis [46].”
Comments to the Author
- R&D (L244-245): I suggest changing “methicillin-resistant S. aureus (MRS), P. aeruginosaand M. intracellulare with MIC values between” to “MRSA, P. aeruginosa and Mycobacterium intracellulare with MIC values ranging between”.
Answer to reviewer 1: This was corrected in the revised manuscript following the reviewer’s comment.
Comments to the Author
- R&D - Table 3: Data for A. baumannii should be aligned in the same row.
Answer to reviewer 1: This was corrected in the revised manuscript following the reviewer’s comment. Moreover, borderlines were marked in Table 3 in order to avoid confusion with the displayed data.
Comments to the Author
- R&D (L310-319): I suggest to change the text to: “The crude extract was partitioned using the modified Kupchan procedure [59] to afford a very active dichloromethane fraction (FD), with MICs of 2-32 mg/L (Table 4), which was submitted to Solid Phase Extraction (SPE) using an RP-18 cartridge to give seven subfractions (R1-R7). RP-HPLC purification of R2 which appeared as the most pure fraction (based on its 1H NMR spectrum), with a MIC value of 4-8 mg/L against S. aureus (Table 5), led to the isolation of a pure compound with a retention time of 35.2 min (see RP-HPLC chromatogram in Figure S4 in the SM). Its HRESIMS spectrum, showing the [M + H]+ ion adduct at m/z 422.3266, the 1H and 13C NMR spectral data (see Figures S5-7 in the SM), and its optical rotation value matched with those of (-)-agelasine B, isolated from an unidentified sponge of the Agelas genus collected in the Okinawan sea. [60,61]”.
Answer to reviewer 1: This was corrected in the revised manuscript following the reviewer’s comment.
Comments to the Author
- R&D (L334-339): I suggest to change the text to: “The antimicrobial activity (MIC) of (-)-agelasine B has been reported against Saccharomyces cerevisiae ATCC 188224 (10 μg/mL) [62], Staphylococcus epidermidis 13889 (1.56 μg/mL), E. faecalis 12964 (6.25 μg/mL), E. faecium 12367 (3.13 μg/mL), S. aureus 12732 (0.78 μg/mL), MRSA (3.13 μg/mL), Candida albicans IFO-1269 (>12.5 μg/mL) and Cryptococcus neoformans TIMM-0354 (3.13 μg/mL) [63], Mycobacterium smegmatis (3.13 μg/mL) [64,65], M. bovis (6.25-12.5 μg/mL) [64], and Proteusbacillus vulgaris (18.75 μg/mL) [66].”
Answer to reviewer 1: This was corrected in the revised manuscript following the reviewer’s comment.
Comments to the Author
- R&D (MAJOR): The authors should check their own MIC-data throughout the R&D section. The MIC-data are sometimes listed as mg/L and sometimes listed as mg/mL, both in tables and text. For instance, in R&D, Line 311 MICs are listed in mg/L and when referring to Table 4, the same values are presents as mg/mL! That is a thousand-fold difference. In addition, when citing literature they use μg/mL for comparison of data. I suggest that they use a common unit of measurement (like μg/mL) throughout the paper.
Answer to reviewer 1:
As we have previously commented, all MICs shown in the study were quantified in mg/L, following the recommendations of international organizations such as EUCAST (European Committee on Antimicrobial Susceptibility Testing). The displayed MICs in mg/mL were typographical errors and in consequence, the entire manuscript has been homogenized with the MIC values in mg/L.
Comments to the Author
- M&M (L403-404): The authors write: “Liquid-liquid fractionation of 6.0 g between CH3OH/CH2Cl2 (1:1 v/v) gave aqueous (WW) and an organic fraction. The WW fraction was extracted with n-butanol (200 mL) to yield the n-butanol fraction (WB = 756.0 mg).” How do you obtain an aqueous fraction from the CH3OH/CH2Cl2 mixture? Should it be H2O/CH2Cl2(1:1 v/v)? Please clarify, and also add yield of the WW fraction.
Answer to reviewer 1: The reviewer is right, the liquid-liquid fractionation was carried out between H2O/CH2Cl2(1:1 v/v) and this mistake was corrected in the revised manuscript. WW is referred to the final aqueous fraction, after extraction of the aqueous phase with n-BuOH, and this nomenclature was also corrected in the revised version of the manuscript. Furthermore, the yield of the WW fraction was added.
Comments to the Author
- M&M (L407): Add space between 400 and mL.
Answer to reviewer 1: This was corrected in the revised manuscript following the reviewer’s comment.
Comments to the Author
- M&M (L417-421): I suggest to change the text to: “The active fraction R2 (181.3 mg) was further fractioned by HPLC. The mobile phase consisted of (A) H2O with 0.04% of trifluroacetic acid, and (B) CH3OH with 0.04% of trifluoracetic acid, and the analysis were run at a flow rate of 2.0 mL/min. A combination of gradient and isocratic elution was used, starting with 30% B, increasing to 100% of B in 30 min, followed by 10 min isocratic at 100% of B.”.
Answer to reviewer 1: This was corrected in the revised manuscript following the reviewer’s comment.
Comments to the Author
- M&M (L469): I suggest changing “MICs” to “MIC”.
Answer to reviewer 1: This was corrected in the revised manuscript following the reviewer’s comment.
Comments to the Author
- M&M (L470): I suggest deleting «the”.
Answer to reviewer 1: This was corrected in the revised manuscript following the reviewer’s comment.
Comments to the Author
- Conclusion (L485): Use abbreviated names for Amphimedon compressa and Haliclona.
Answer to reviewer 1: This was corrected in the revised manuscript following the reviewer’s comment.
Comments to the Author
- Conclusion (L488): Change “estudies” to “studies”.
Answer to reviewer 1: This was corrected in the revised manuscript following the reviewer’s comment.
Reviewer 2 Report
The revised version of the paper addresses the reviewer's comments. Unfortunately, the similarity of the stuctures of amphitoxins and halitoxins hampered their separation and the possibility to test the activity of the pure compounds.
Author Response
Reviewer 2
The revised version of the paper addresses the reviewer's comments. Unfortunately, the similarity of the stuctures of amphitoxins and halitoxins hampered their separation and the possibility to test the activity of the pure compounds.
Answer to reviewer 2: Thank you for the efforts of reviewer 2 to revise the manuscript again.
This manuscript is a resubmission of an earlier submission. The following is a list of the peer review reports and author responses from that submission.
Round 1
Reviewer 1 Report
Briefly, this manuscript describes the extraction and antibacterial screening of 51 sponges and 13 ascidians, collected on the coast of the Yucatan Peninsula in Mexico. The extracts were screened against 4 bacterial strains using broth microdilution assay. Extracts of 8 of the species, all sponges, displayed activity against some of the pathogens tested. From the sponge Amphimedon compressa, a mixture of known halitoxins and amphitoxins were purified and characterized. The results indicate that the variety of species found the coast of Yucatan Peninsula is a potential source for discovering novel antibacterial drug leads.
I find this work of good quality, whose experimental design is sound. The data is in general well-presented and discussed, however, there is scope for improvement in certain portions of the text and figures/tables.
For practical reasons, I will list my comments in a chronological order. Some comments are very minor (grammatical faults) while others are more major. For the latter, I will add (MAJOR) to highlight their importance.
- Abstract (L16): I suggest inserting “A total of” upfront of “51 sponges ….”
- Abstract (L17): I suggest changing “The resulting extracts were evaluated against” to “The resulting extracts were screened for antibacterial activity against”.
- Abstract (L18) (MAJOR): The authors claim the extracts were tested for activity against multidrug-resistant (MDR) bacterial pathogens (also in L26, L292, L313). However, the bacterial strains listed in M&M (L390-L396) are not presented as drug-resistant strains at all. Just because the selected bacterial strains form part of the ESKAPE group, they are not necessarily drug-resistant. This should be clarified and corrected. Information about the resistance profile of the strains should be added (for instance in table 6).
- Abstract (L18-L27): Species names should be in Italic font.
- Abstract (L21): I suggest changing “determined by broth microdilution assay” to “determined using a broth microdilution assay”.
- Abstract (L25-L27): I suggest to split the following sentence into two sentences; from “It is also the first report of the antimicrobial activity of halitoxins and amphitoxins against major multidrug-resistant human pathogens which were isolated by bioguided fractionation of the active extracts from the sponge Amphimedon compressa” to “It is also the first report of the antimicrobial activity of halitoxins and amphitoxins against major multidrug-resistant human pathogens. The compounds were isolated by bioassay-guided fractionation of the active extracts from the sponge Amphimedon compressa”.
- Introduction (L76): I suggest changing”Oceans …” to “The oceans …”.
- Introduction (L80): I suggest changing “natural marine products” to “marine natural products”, which is a more common phrase for such compounds.
- R&D (L128): I suggest changing “…were included in this study…” to “…were used as test strains in this study…”
- R&D (L128): I suggest deleting «realistic».
- R&D (L134-L143): I suggest deleting this section. I do not think you have to argue for using MIC-assays in your screening. It is common procedure. Furthermore, susceptibility breakpoints determined using the EUCAST/CLSI-protocols are for more of interest when testing pure compounds. This study is dealing with extracts and semipure fractions.
- R&D (L148): I suggest changing “Of the 64 extracts, 9 displayed” to “Of the 64 extracts tested, 9 displayed”.
- R&D (L149-L153): I suggest splitting the sentence into two sentences.
- R&D (L154): I suggest changing “the four bacteria tested” to “all the four bacterial strains tested”.
- R&D and Conclusion (the whole section): Only write the full species names (bacteria and invertebrates) first time mentioned. Otherwise abbreviate. Ex: Agelas citrina (first time), A. citrina (next time).
- Table 1: I suggest listing the species after phyla, order, and family (at least phyla and order) as in Table 5. Now it looks random, not even alphabetical. The Code-column could be deleted if there is need for extra space. Alternatively, the code could be added behind each species. Ex: Clavelina sp. (T18-M1).
- R&D (L171-L180)(related to comment no. 11): It is more relevant to compare the extract against each other, than comparing to commercial antibiotics. Extracts are not “realistic therapies”. I suggest revising this section.
- R&D (L188, L190, L300, L311): In vitro should be in italic.
- R&D (L238-L282) (MAJOR): Section 2.2; Why do the authors present the isolation and characterization of already known compounds, even known from the same species? And why not prioritize analyzing the extract from Agelas citrina or Haliclona (Rhizoniera) curacoensis, which both are more potent against the test strains? Please clarify. I suggest moving Fig 1, Fig 2, and Fig 3 to supporting info.
- R&D (L274-L276): I suggest moving this section (starting with “Subfraction R4H2 …”) to L251, after “(Figure 1)”.
- R&D (L287-L291): I suggest to have a look at the publication by Turk et al., 2008: https://doi.org/10.1016/S1572-5995(08)80009-9 and references therein.
- M&M (L348): I suggest changing “Bioguided isolation of …” to “Bioassay-guided isolation and characterization of …”.
- M&M (L354): I suppose “R6: 192.3 g” should be “R6: 192.3 mg”. Please check.
- M&M (L357): I suggest changing “in an Agilent 1100” to “using an Agilent 1100”.
- M&M (Table 5, Page 14-19)(MAJOR): I suggest to either: i) combine this table with Table 1, or ii) move this table to supporting info. The “Phylum: Chordata” and “Order: Aplousobranchia” should be inserted below row 1 in the table. The only results in this table is the extract yield for each species, and the yield does not really give any valuable information since the original amount of material subjected for extraction is not provided. The most valuable information in this table is the collection sites.
- M&M (Table 6, Page 20). Unless the drug-resistance profile of the strains can be added to the table, I suggest deleting or moving it to Supporting Info.
- M&M (L389-390): I suggest changing “The bacterial strains used to study the antibacterial activity of the crude extracts as well as their main characteristics are shown in Table 6. The bacteria studied were the Gram-negative …” to “The bacterial strains used to study the antibacterial activity of the crude extracts were the Gram-negative …”
- General: Numbering of Figures and Tables should be corrected (There are now two “Figure 1´s” and no “Table 2”) in the manuscript.
- General (MAJOR): Regarding literature search; the authors should check for previous names and synonymes (in taxonomic databases), at least for the “active species”. Ex: Aiolochroia crassa: This sponge has been reported under different names, such as Suberea crassa (Hyatt, 1875), Aiolochroia ianthella (De Laubenfels, 1949), Ianthella ianthella (De Laubenfels, 1949) and Ianthella ardis (De Laubenfels, 1950). Previous studies on the sponge A. crassa (and also A. compressa!) have led to the discovery of antimicrobial activity of its extracts (https://doi.org/10.3354/ame040191), and the isolation and identification of several bromotyrosine derivatives: N-methylaerophobin-2, aerophobin-1, aerophobin-2, purealidin L, isofistularin-3, araplysillin III, hexadellin C, N,N,N-trimethyl-3-bromo-4-O-methyltyrosine and N,N,N-trimethyl-3-bromotyrosine (https://doi.org/10.1016/S0040-4020(99)00553-0 and https://doi.org/10.1515/znc-1998-5-615). Aeroplysinin 1 and ianthelline were reported from the sponge Ianthella ardis (https://doi.org/10.1016/S0040-4039(00)84977-1).
Reviewer 2 Report
This study represents the first report of antibacterial activity for a variety of sponges (51) and ascidians (13) collected on the coast of the Yucatan Peninsula.
The organic extracts were evaluated for their in vitro antimicrobial activity against four species of multidrug-resistant bacterial pathogens. Only the extracts from eight sponges were active, showing some of these minimum inhibitory concentration values similar to those exhibited by known antibiotics.
Active extract from the sponge Amphimedon compressa was further purified by bioguided fractionation to obtain the subfraction R4H2 which was characterized by 1H and 13C NMR spectroscopy as a mixture of amphitoxins and halitoxins, bioactive compounds already identified many decades ago from sponges.
Major comments:
Although a wide survey of 64 marine organisms as potential source of antibacterial compounds was performed, the study shows some weakness.
1) The results on the antibacterial activity of the organic extracts from some sponges are preliminary until the pure compounds responsible of this bioactivity are isolated.
2) In this study no new compounds were identified. Only in a subfraction from the sponge Amphimedon compressa a mixture of amphitoxins and halitoxins was characterized. These compounds, already identified in sponges and known for their antibacterial activity, were shown in this paper to be also active against major multidrug-resistant human pathogens.
3) A comparison of the antimicrobial activity of amphitoxins and halitoxins against multidrug-resistant bacterial pathogens should be examined.
4) The title "Marine organisms from the Yucatan Peninsula (Mexico) as a potential natural source of new antibacterial compounds" should be changed to "Marine organisms from the Yucatan Peninsula (Mexico) as a potential natural source of antibacterial compounds" to better highlight the results presented in the paper.
Minor comments
1) In Materials and Methods there are two sections with the same number: 3.3 Antimicrobial activity assays and 3.3 Bioguided isolation of the Amphimedon compressa crude extract. The two sections are reported in the text before and after Table 5 and Table 6, respectively.
2) The text lines 131-143 reported in Results and Discussion should be moved in a more general part, e.g. in the Introduction.